# Stronger Increases in Cognitive Functions among Socio-Economically Disadvantaged Older Adults in China: A Longitudinal Analysis with Multiple Birth Cohorts

**DOI:** 10.3390/ijerph17072418

**Published:** 2020-04-02

**Authors:** Fan Yang, Jiangling Cao, Dongfu Qian, Aixia Ma

**Affiliations:** 1School of International Pharmaceutical Business, China Pharmaceutical University, No.639 Longmian Str, Jiangning District, Nanjing 211198, China; youngfan@njmu.edu.cn; 2School of Health Policy & Management, Nanjing Medical University, 101 Longmian Avenue, Nanjing 210029, China; caojiangling1224@163.com (J.C.); dongfu016@126.com (D.Q.)

**Keywords:** Cognitive decline, Socio-economically disadvantaged, Cohort trend, Socio-economic disparity

## Abstract

Highly variable changes in cognitive functions occur as people get older, and socio-economically disadvantaged older adults are more likely to suffer from cognitive decline. This study aims to identify the longitudinal trend in cognitive functions among different socio-economic groups of older adults. The Chinese Longitudinal Healthy Longevity Survey (CLHLS) followed up 32,323 individuals aged 65 years and older over a 12-year period. A series of mixed-effects models was used to explicitly assess cohort trend and its socio-economic disparity in the cognitive functions of older adults. Scores for significant increase in cognitive functions by birth cohort were smaller by 0.49, 0.28, and 0.64 among older adults with more educational experience, a lower household income, or economic dependence relative to their counterparts. Scores for differences in cognitive function between older adults with higher and lower incomes were smaller by 0.46 among those living in urban areas than among those living in rural areas. Although there were larger cohort growth trends in cognitive functions among older adults with lower educational attainment, lower household income, and who were economically dependent, effective public intervention targeting these socio-economically disadvantaged populations is still necessary.

## 1. Introduction

China has rapidly become an aging society, where the number of people aged 65 and older will increase from 6.9% of the population in 2000 to 22.7% in 2050 [1,2]. With such rapid growth in the older population, cognitive decline among older adults could bring a significant future burden to all of society [3]. Cognitive aging has been defined as “a process of gradual, ongoing, yet highly variable changes in cognitive functions that occur as people get older” [4]. Cognitive decline might contribute to restricted independence, a decline in the quality of life, exacerbation of depression, and increased premature mortality among older adults [5,6,7]. Socio-economic factors (such as education and income levels) may affect the risk of cognitive decline among older adults [8,9,10,11,12,13,14,15], in line with the suggested socio-economic gradient in health.

In recent decades, China promoted equality and conducted industrialization, which may bring different changes in cognitive function among socio-economically disadvantaged older adults. For example, the government health-care subsidies, and the health-care system were consistently improved in poor areas in China [10,15]. There were considerable improvements in nutritional intake [11] and the accessibility and utilization of the health-care service [10] among disadvantaged socio-economic groups. These improvements may have contributed to more favorable conditions for the prevention and treatment (than previously) of cognitive decline. To gain deeper insights into the development of cognitive functions, it is necessary to study the longitudinal trend in cognitive functions among different socio-economic older adults. Also, current research has indicated both gender [3] and area differences [16] in cognitive function, which might interfere with the impact of individual socio-economic status. Nearly two-thirds of the current older adults in China were involved in farming and agricultural production when they were young, preventing them from having a stable income in their later life [17]. Economic independence may reflect the financial situation and income sources of older adults in China [15,18]. Therefore, it is essential to determine the socio-economic effects (such as economic independence) on cognitive functions among older adults of different genders and living in different areas.

However, some studies showed that older adults with lower socio-economic status have weaker cognitive functions. Most studies were cross-sectional and did not ascertain the longitudinal change and the cohort effects on cognitive function during the aging process. A longitudinal study with multiple birth cohorts has the clear advantage of exploring complex longitudinal trends and cohort effects in cognitive functions during the aging process and providing precise explanations of socio-economic effects [11,12,13,14]. Many research studies assessing longitudinal trends in cognitive functions came from developed countries, particularly the United States. Moreover, owing to data limitations, most of the studies have typically investigated subjects younger than 80 years. Accordingly, our study used the longitudinal data of the Chinese Longitudinal Healthy Longevity Survey (CLHLS) [15,17,19,20] to identify the longitudinal trend in cognitive functions among different socio-economic older adults. The CLHLS was the first nationwide longitudinal survey in China, concentrating on the most extensive sample of older adults aged 80 and above.

## 2. Methods 

### 2.1. Study Samples

The CLHLS [15,17,19,20] was the first nationwide longitudinal survey conducted in China, with a long-term follow-up period from 1998 to 2014. The CLHLS involved one of the most extensive samples of older adults from 631 counties/cities in 22 provinces (China has a total of 31 provincial-level regions), covering up to 85% of the total Chinese population in China. Survey protocols, instruments, and the process for obtaining informed consent for this study were approved by Duke University Health System’s Institutional Review Board (IRB00001052–13074). The CLHLS began enrolling individuals in 1998 and carried out the subsequent six rounds of data collection in 2000, 2002, 2005, 2008, 2011–2012, and 2014.

The CLHLS began to establish the sampling frame with available centenarians from sampled counties/cities. Multiple effective ways were used to verify the age of each centenarian, including birth certificates, genealogical documents, household booklets, and the ages of the subjects’ children and siblings. Next, the CLHLS randomly selected older adults aged 80–89 years and aged 90–99 years from a nearby geographical unit to match each centenarian, with pre-determined age and gender. The age of each older adult aged 80–89 years and aged 90–99 years depended on the last digit of the random code assigned to the centenarian. An even number indicated that the interviewee was female; an odd number indicated a male interviewee. The nearby geographical unit meant the same village or street, if applicable, or the same town, county, or city. The CLHLS designed the sampling strategy to guarantee comparable numbers of males and females aged from 80 to 99 years. A younger cohort aged from 65 to 79 was followed as a comparison group in 2002, using a similar sampling strategy. More details of the sampling design and data assessment of the CLHLS are available elsewhere [17,19,20].

We used data from the CLHLS in 2002, 2005, 2008, 2011–2012, and 2014, as the surveys in 1998 and 2000 did not collect data on older adults aged 65 to 79 years. The total sample for analysis contained 56,565 observations on 32,323 individuals.

### 2.2. Variable Definitions

Cognitive function was measured using the Chinese version of the Mini Mental State Examination (MMSE) adapted from the scale developed by Folstein and colleagues [21,22], evaluating five aspects of functioning: orientation, reaction, attention and calculation, recall, and language [15]. The Chinese version of the MMSE we used fits the cultural and socio-economic background in China, and the reliability was high (Cronbach’s a = 0.94) for the current samples of Chinese older adults [15]. This MMSE consists of 24 questions covering orientation (six questions), reaction (three questions), attention and calculation (six questions), recall (three questions), and language (six questions) [15,23]. All the questions in the MMSE must be answered by the respondents themselves. There are three possible answers for each question: correct, wrong, and unable to answer. Considering the evaluation of alternative ways for dealing with responses of “unable to answer” in previous literature, we coded unanswerable responses as incorrect answers [15]. Except for the question asking participants to name types of food in 1 minute, which has a possible score of 7, each question has a score of 1 for a correct answer and a score of 0 for an incorrect answer [15,23]. The MMSE has a total score of 30, and lower scores mean poorer functioning. Education was closely correlated with MMSE scores [15,23]. A large proportion (60.3%) of the older adults in this study had no formal schooling. Therefore, in this study, the education-based MMSE cut-off point was used to define cognitive impairment. This cut-off-point is widely accepted for screening for cognitive impairment in the older adult population with less education [23]. An MMSE score <18 for participants with no formal education had, an MMSE score of <21 for those with 1 to 6 years of education had, or an MMSE score of <25 for those with more than 6 years of education was defined as indicating cognitive impairment [23].

There were five consecutive birth cohorts: cohort 1891–1900, cohort 1901–1910, cohort 1911–1920, cohort 1921–1930, cohort 1931–1940, and cohort 1941–1950. We classified gender as being either “male” or “female,” and current residence as being in either a “rural area” or an “urban area [24].” The data on education were classified as either “at least one year of schooling” or “no schooling,” as most of the older Chinese adults had the little educational experience, and other alternative coding did not improve the estimates [18]. Marital status was classified as married, widowed, divorced/separated, or never married. If the income per capita of a household last year was more than 100,000 Chinese yuan, the CLHLS coded 99998. Household income was classified based on the median of per capita net income for each survey year: “high income” or “low income.” This study did not use household income as a continuous variable because the CLHLS did not code the exact value of income above 100,000 Chinese yuan. Economic independence is also a socio-economic factor that may reflect the economic conditions and sources among the Chinese older population because nearly two-thirds of the Chinese older population do not have a stable income (or pension) in their later life [17,18]. Economic independence was classified into two categories: “economically independent” (relying mainly on a pension or one’s own financial resources) or “economically dependent” (relying mainly on a spouse, children, or other family members, government subsidy, or other sources) [18]. Using the World Health Organization standards, smoking data were divided into currently smoking, past smoking, or non-smoking, according to the subjects’ having smoked 100 cigarettes over a lifetime and currently or in the past smoking cigarettes [25]. Alcohol intake data were divided into currently drinking alcohol, past drinking, or non-drinking, according to the subjects’ alcohol intake being more than once per month during the past 12 months and current or past drinking of alcohol [25]. Frequency of fresh fruit intake was stratified into four groups: eating it every day or almost every day, quite often, occasionally, and rarely or never. Frequency of fresh vegetable intake was also stratified into four groups: eating it every day or almost every day, quite often, occasionally, and rarely or never. Functional disability was assessed by self-report using the Katz questionnaire for basic activities of daily living (ADL). In the clinical and epidemiological fields, this evaluation tool is used to assess functional independence [19]. ADL was evaluated by six daily self-care tasks: bathing, eating, getting in and out of bed, dressing, going to the toilet, and maintaining control of urine and feces. Respondents who reported any assistance in any items were categorized as ADL disability, whereas respondents who completed all items without any help were categorized as independent of ADL disability [19]. The trained investigator used a mercury sphygmomanometer (upper arm type; Yuyue, Jiangsu, China) to measure participants’ blood pressure after they had rested for at least five minutes. Korotkoff phase I was designated for the systolic blood pressure (SBP) values (mmHg) [26]. The SBP value was a continuous variable.

### 2.3. Statistical Analysis

Linear mixed-effects models are especially useful in longitudinal studies [27] and were used to identify cohort trends in cognitive functions among different socio-economic older adults. The socio-economic indicators were added to the models to identify socio-economic disparities in cognitive functions of older Chinese adults of different genders and areas of residence. The models can be formulated as follows [27]:
(1)$$\begin{array}{cc} Y_{it}= & \left(\beta_{0}+\mu_{0i}\right)+\left(\beta_{1}+\mu_{1i}\right)Birth\;cohort_{it}+\left(\beta_{2}+\mu_{2i}\right)Gender_{it}+\left(\beta_{3}+\mu_{3i}\right)Areas_{it}\\ & +\left(\beta_{4}+\mu_{4i}\right)Birth\;cohort_{it}*Gender_{it}+\left(\beta_{5}+\mu_{5i}\right)Birth\;cohort_{it}*Areas_{it}\\ & +\;\left(\beta_{6}+\mu_{6i}\right)Adjusted\;variables_{it}+e_{it} \end{array}$$
(2)$$\begin{array}{cc} Y_{it}= & \left(\beta_{0}+\mu_{0i}\right)+\left(\beta_{1}+\mu_{1i}\right)Birth\;cohort_{it}+\left(\beta_{2}+\mu_{2i}\right)Gender_{it}+\left(\beta_{3}+\mu_{3i}\right)Areas_{it}\\ & +\left(\beta_{4}+\mu_{4i}\right)Birth\;cohort_{it}*Gender_{it}+\left(\beta_{5}+\mu_{5i}\right)Birth\;cohort_{it}*Areas_{it}\\ & +\left(\beta_{6}+\mu_{6i}\right)Education_{it}+\left(\beta_{7}+\mu_{7i}\right)Household\;income_{it}\\ & +\left(\beta_{8}+\mu_{8i}\right)Economic\;independence_{it}+\left(\beta_{9}+\mu_{9i}\right)Adjusted\;variables_{it}+e_{it} \end{array}$$
(3)$$\begin{array}{cc} Y_{it}= & \left(\beta_{0}+\mu_{0i}\right)+\left(\beta_{1}+\mu_{1i}\right)Birth\;cohort_{it}+\left(\beta_{2}+\mu_{2i}\right)Gender_{it}+\left(\beta_{3}+\mu_{3i}\right)Areas_{it}\\ & +\left(\beta_{4}+\mu_{4i}\right)Birth\;cohort_{it}*Gender_{it}+\left(\beta_{5}+\mu_{5i}\right)Birth\;cohort_{it}*Areas_{it}\\ & +\left(\beta_{6}+\mu_{6i}\right)Education_{it}+\left(\beta_{7}+\mu_{7i}\right)Household\;income_{it}\\ & +\left(\beta_{8}+\mu_{8i}\right)Economic\;independence_{it}\\ & +\left(\beta_{9}+\mu_{9i}\right)Economic\;independence_{it}*Gender_{it}\\ & +\left(\beta_{10}+\mu_{10i}\right)Education_{it}*Gender_{it}+\left(\beta_{11}+\mu_{11i}\right)Household\;income_{it}*Gender_{it}\\ & +\left(\beta_{12}+\mu_{12i}\right)Economic\;independence_{it}*Areas_{it}\\ & +\left(\beta_{13}+\mu_{13i}\right)Education_{it}*Areas_{it}+\left(\beta_{14}+\mu_{14i}\right)Household\;income_{it}*Areas_{it}\\ & +\left(\beta_{15}+\mu_{15i}\right)Adjusted\;variables_{it}+e_{it} \end{array}$$
(4)$$\begin{array}{cc} Y_{it}= & \left(\beta_{0}+\mu_{0i}\right)+\left(\beta_{1}+\mu_{1i}\right)Birth\;cohort_{it}+\left(\beta_{2}+\mu_{2i}\right)Gender_{it}+\left(\beta_{3}+\mu_{3i}\right)Areas_{it}\\ & +\left(\beta_{4}+\mu_{4i}\right)Birth\;cohort_{it}*Gender_{it}+\left(\beta_{5}+\mu_{5i}\right)Birth\;cohort_{it}*Areas_{it}\\ & +\left(\beta_{6}+\mu_{6i}\right)Education_{it}+\left(\beta_{7}+\mu_{7i}\right)Household\;income_{it}\\ & +\left(\beta_{8}+\mu_{8i}\right)Economic\;independence_{it}+\left(\beta_{9}+\mu_{9i}\right)Economic\;independence_{it}*Gender_{it}\\ & +\left(\beta_{10}+\mu_{10i}\right)Education_{it}*Gender_{it}+\left(\beta_{11}+\mu_{11i}\right)Household\;income_{it}*Gender_{it}\\ & +\left(\beta_{12}+\mu_{12i}\right)Economic\;independence_{it}*Areas_{it}+\left(\beta_{13}+\mu_{13i}\right)Education_{it}*Areas_{it}\\ & +\left(\beta_{14}+\mu_{14i}\right)Household\;income_{it}*Areas_{it}\\ & +\left(\beta_{15}+\mu_{15i}\right)Birth\;cohort_{it}*Economic\;independence_{it}\\ & +\left(\beta_{16}+\mu_{16i}\right)Birth\;cohort_{it}*Education_{it}\\ & +\left(\beta_{17}+\mu_{17i}\right)Birth\;cohort_{it}*Household\;income_{it}+\left(\beta_{18}+\mu_{18i}\right)Adjusted\;variables_{it}+e_{it} \end{array}$$
where $Y_{it}$ is the response vector for individual *i* at time *t* (the measurement instance for cognitive functions);$\;\mu_{0i}$,$\;\mu_{1i}$..., $\mu_{ki}$ are the differences between β_0,_
β_1_ ..., β_k_ and the intercept or slopes of subject *i* (random effects); β_0,_
β_1_ ..., β_k_ are the subjects’ mean intercept or slopes for the above explanatory variables (fixed effects); $e_{it}$ is the random error within subjects over time. Age, marital status, smoking, drinking, frequency of fruit intake, frequency of vegetables intake, SBP, and ADL disability were adjusted variables in these different models, respectively.

In Equation (1), the coefficients for birth cohorts can be interpreted as a cohort trend, and the coefficients for birth cohort and gender (or area) interaction terms can be interpreted as differences in the experienced period trend between males and females (or between urban areas and rural areas). In Equation (2), the coefficients for economic independence, household income, and education can be interpreted as socio-economic disparities. In Equation (3), the coefficients for economic independence (or household income or education) and gender (or area) interaction terms can be interpreted as differences in this socio-economic disparity between males and females (or between urban areas and rural areas). In Equation (4), the coefficients for birth cohort and education (or household income or economic independence) interaction terms can be interpreted as differences in the experienced cohort trend between older adults having at least one year of schooling and no schooling.

All statistical tests were 2-sided and were performed using Stata version 12 (StataCorp LLC: College Station, TX, USA).

## 3. Results

The demographics and socio-economic characteristics of the sample set are presented in Table 1. The scores for cognitive functions by different education, household income, and economic dependence across age and cohorts are shown in Table 2, Table 3 and Table 4. The prevalence of cognitive impairment (%) by different education, household income, and economic dependence are also shown across age and cohorts in Figure 1.

Cognitive functions significantly increased with birth cohort (Table 5, unadjusted model A). When other socio-economic factors were adjusted, cognitive functions also increased significantly by 0.28 scores per birth cohort (coefficient [95% confidence intervals]: 0.28 scores [0.14, 0.43]; Table 5, adjusted model A). Older male adults and those living in urban areas had 4.16 and 1.71 scores higher in cognitive functions relative to females (coefficient [95% CI]: 4.16 scores [3.74, 4.58]) and those living in rural areas (coefficient [95% CI]:1.71 scores [1.35, 2.08]), respectively. The interaction between birth cohorts and gender, and the interaction between birth cohorts and area, were indicators that the significant increase in cognitive functions per cohort was 0.69 scores smaller in women than in men (coefficient [95% CI]: −0.69 scores [−0.80, −0.57]), and was 0.21 scores smaller among those who lived in rural areas than those living in urban areas (coefficient [95% CI]: −0.21 scores [−0.31, −0.11]).

Older adults with higher educational attainment, higher household income, and economic independence scored 1.38, 0.85 and 0.54 higher for cognitive functions compared to their counterparts (coefficient [95% CI]: 1.38[1.21, 1.55], 0.85[0.72, 0.98] and 0.54 scores [0.38, 0.71]; Table 5, adjusted model B), respectively. The magnitude of differences in cognitive functions between older adults with higher and lower incomes was reflected in scores lower by 0.46 among those living in urban areas than among those living in rural areas (coefficient [95% CI]: −0.46 scores [−0.71, −0.20]; the interaction between household income and area in Table 5, adjusted model C).

The interaction between birth cohorts and gender, and the interaction between birth cohorts and area, were indicators that the significant increase in cognitive functions by birth cohort scored 0.49 less among older, less well-educated adults with (no schooling) relative to those with more educational experience (at least one year of schooling) (coefficient [95% CI]: −0.49 scores [−0.62, −0.35]; the interaction between birth cohorts and household income in Table 5, adjusted model D). Compared with those with higher household income, and who were economically independent, the significant increase in cognitive functions per birth cohort scored 0.28 and 0.64 less among older adults with a lower household income and who were economically dependent (coefficient [95% CI]: −0.28 scores [−0.39, −0.35] and −0.64 scores [−0.78, −0.50]; Table 5, adjusted model D), respectively.

## 4. Discussion

In our study, the follow up of multiple cohorts with panel updates over time was advantageous for assessing the longitudinal trend among different socio-economic older adults and distinguishing socio-economic disparities in cognitive functions among older adults of different genders and areas of residence. The significant increase in cognitive functions per birth cohort was stronger among older Chinese adults with less education, lower household income, and who were economically dependent. Furthermore, the magnitude of differences in cognitive function between older adults with higher and lower incomes was larger among those living in rural areas than among those living in urban areas.

In the United States [11,14], the prevalence of cognitive decline was also higher in older cohorts than in younger cohorts. Also, several cross-sectional research studies have examined gender [3] and area of residence differences [16] in the cognitive decline of older adults. To our knowledge, our longitudinal analysis is the first to identify the cohort trends on cognitive functions of older adults with different socio-economic status in China, showing a stronger increase per birth cohort in cognitive functions among older adults with less educational attainment, lower household income, and who were economically dependent.

Exposure to different environments, having different health behaviors and chronic diseases, and utilization of the health-care service among people of different socio-economic status can lead to a potential social gradient in cognitive function in later life [10,15]. Education was a strong predictor for cognitive functions in prior research [5,7,10,13]. Older adults with higher household incomes and economic independence also had higher cognitive functions. However, the promotion of equality, as well as development of industrialization, in recent decades not only brought a great improvement in the levels of education and income [28] but also a general improvement in nutritional intake [11] and utilization of the health-care service [10,15,29]. These factors may be responsible for the stronger cohort increase for cognitive functions among these socio-economically vulnerable older adults. For example, the proportion of food protein increased continuously in China from 1997 to 2009, particularly among low-income residents [30,31]; and the growth rate of meat protein intake among low-income residents was higher than that of high-income residents [30,31].

Also, research has shown that a gatekeeper mechanism implemented in China’s poor areas improved the distribution of government health-care subsidies [32], and some health services have improved due to this equitable distribution of subsidies [33]. The consistent improvement of the health-care system promotes the accessibility of medical services, especially among these vulnerable groups. These improvements have also contributed to more favorable conditions for prevention and treatment (than previously) of cognitive decline among socio-economically vulnerable old adults. Overall, this positive cohort growth trend for cognitive functions may be beneficial considering the rapid growth of the older population in China [2], and may also have caused the socio-economic disparities in cognitive functions to narrow among the younger generations. The higher scores for cognitive functions among younger generations could help to prevent loss of independence, decline in quality of life, exacerbation of depression, incidence of dementia, and mortality among older adults [5,6,7,15].

Although old adults living in urban areas had higher cognitive functions relative to those living in rural areas, our longitudinal analysis also revealed that the magnitude of the difference in cognitive functions between the older adults with higher and lower incomes was larger among those living in rural areas. China has a unique urban–rural structure where economic development and stratification systems differ from those in western developed countries, as nearly two-thirds of the current old adults in China were involved in farming and agricultural production when they were young [15,17,34,35,36]. Older adults in rural areas were generally socio-economically disadvantaged with limited social participation and less cognitively stimulating environments relative to those in urban areas [13,14,33,37]. Rural residents with lower income levels may have lower resources and opportunities for social engagement and interaction to prevent their cognitive decline than their counterparts. Rural areas also lack economic resources. Deprived neighborhoods may result in fewer resources for health-service utilization [11,12,13,14,16,34], which otherwise may have posed barriers to against cognitive decline among impoverished rural residents [38]. It may be one reason for our finding of a more significant magnitude of income-related disparity in cognitive functions among rural-area older adults in China.

There were some limitations to this study. First, in the process of the investigation, it was not possible to follow up on some interviewees, and the attrition rates between each wave were high because the older respondents themselves must answer the MMSE, and some interviewees had died. Older adults with better cognitive functions were further included in the study and this led to possible survival bias because those interviewees with poor cognitive functions had died and were excluded from the study. In addition, there may also be a survivor bias in the data because older adults with poor cognitive functions may have died before the cohort was investigated. Second, subsequent longitudinal studies should have longer birth cohort spans with a larger number of middle-aged people in order to better explore cohort trends for cognitive functions and the socio-economic factors of individuals. Moreover, although the CLHLS sampling strategy was designed to ensure comparable numbers of randomly selected male and female octogenarians, nonagenarians, and centenarians [15,17,19,20,21], considering the potential selection biases that exist in the complex sampling design survey, sample weights have been released by the CLHLS. We adjusted the variables (such as age, marital status, smoking, drinking, frequency of fruit intake, frequency of vegetable intake, SBP, and ADL disability) in the models during the data analysis. However, some other impact factors, such as physical activity, medical history, and sensorial impairments, were not collected in the CLHLS and cannot be added to these analysis models. Individuals who had been diagnosed with the neurodegenerative disease or dementia were not excluded from the CLHLS. Lastly, the MMSE is inadequate as a sole measure of cognitive impairment because the MMSE has limited diagnostic utility in this setting. 

## 5. Conclusions

There were stronger cohort growth trends in cognitive functions among socio-economically disadvantaged older adults. However, the income-related disparity in cognitive functions was greater among older adults living in rural areas.

## Figures and Tables

**Figure 1 ijerph-17-02418-f001:**
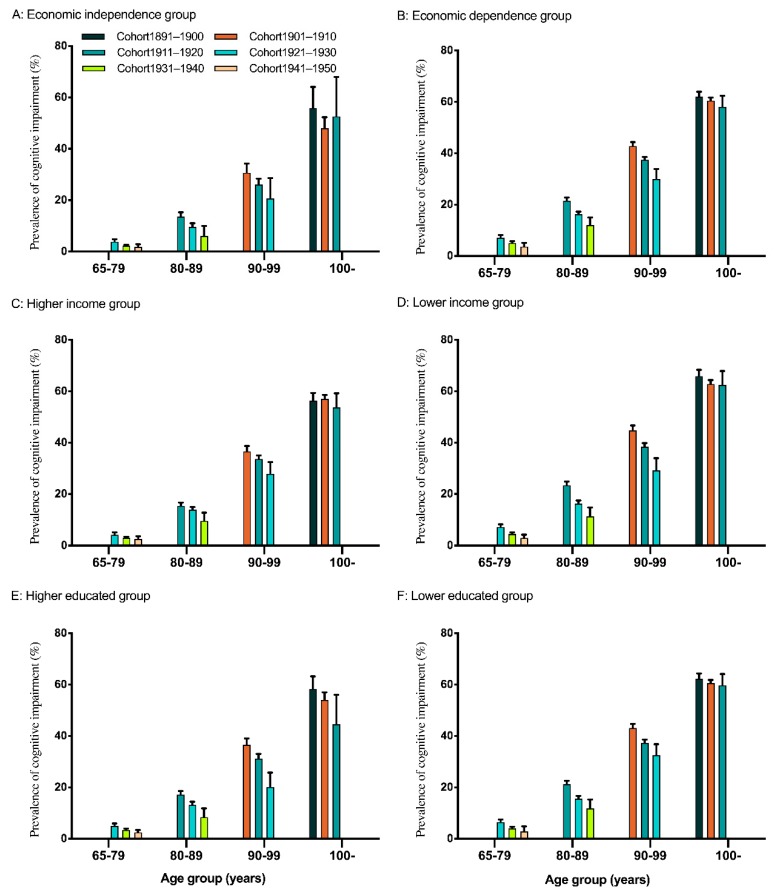
The prevalence of cognitive impairment (%) by different socio-economic status group among older adults.

**Table 1 ijerph-17-02418-t001:** Sample distribution and demographic characteristics of older adults - CLHLS.

Characteristics	2002	2005	2008	2011	2014
Follow up		6087	6236	4482	2974
New participants	14,950	7876	8184	3228	2548
Deceased interviewees/ No response		8863	7727	9938	4736
Surviving interviewees, mean (standard error) or %					
Birth cohorts ^a^					
Cohort 1891–1900	11.8	3.7	0.7	0.1	0.1
Cohort 1901–1910	26.9	25.5	22.4	11.6	5.2
Cohort 1911–1920	27.1	30.6	30.8	25.2	20.1
Cohort 1921–1930	20.5	19.1	22.9	27.9	29.7
Cohort 1931–1940	13.7	21.2	18.6	26.4	32.9
Cohort 1941–1950			4.7	8.8	12.1
Age group (years) ^b^					
65–79	31.3	33.0	27.3	33.2	33.7
80–89	26.2	24.2	25.8	27.5	31.3
90–99	22.9	24.8	27.4	25.5	22.9
100–	19.6	18.0	19.4	13.8	12.1
Men ^c^	42.9	43.3	43.1	45.4	46.4
Urban area ^d^	46.0	45.1	40.0	48.7	47.0
Economic independence ^e^	26.8	29.5	26.2	28.0	27.1
Household income per capita ^f^	48.2	44.7	47.3	50.8	47.3
Education ^g^	38.5	39.8	38.0	42.4	43.1
Marital status ^h^					
Never married	1.2	0.8	0.8	1.0	1.1
Widowed	66.3	65.4	66.5	59.9	58.1
Divorced/separated	2.4	2.3	2.0	1.9	1.8
Married	30.1	31.6	30.7	37.1	39.0
Smoking ^i^					
Currently smoking	65.3	63.4	66.9	64.9	69.3
Past smoking	16.1	17.0	16.0	16.7	13.8
Non-smoking	18.6	19.6	17.1	18.4	16.8
Drinking ^j^					
Currently drinking	66.8	65.3	69.1	67.3	74.6
Past drinking	12.5	14.1	13.9	15.3	10.2
Non-drinking	20.7	20.6	17.1	17.5	15.2
Frequency of fruit intake ^k^					
Every day or almost every day	11.1	11.4	13.0	13.8	14.5
Quite often	22.7	24.9	25.3	24.0	27.0
Occasionally	39.8	36.8	37.3	35.1	33.0
Rarely or never	26.4	26.9	24.3	27.2	25.5
Frequency of vegetables intake ^l^					
Every day or almost every day	52.2	49.8	62.3	56.8	55.5
Quite often	33.7	35.2	25.6	31.3	32.7
Occasionally	10.4	10.9	9.5	8.2	8.2
Rarely or never	3.7	4.0	2.5	3.7	3.6
ADL disability ^m^	29.9	25.1	21.1	26.7	24.6
Cognitive impairment ^n^	25.0	25.7	28.2	23.6	21.4
SBP ^o^	133.8(0.1)	130.9(0.2)	135.8(0.2)	136.7(0.2)	139.6(0.3)
Cognitive functions ^p^	22.1(0.1)	22.0(0.1)	21.2(0.1)	22.7(0.1)	23.2(0.1)

^a^ Birth cohorts was a continuous variable: cohort 1891–1900, cohort 1901–1910, cohort 1911–1920, cohort 1921–1930, cohort 1931–1940, and cohort 1941–1950. ^b^ Age was a continuous variable, the unit is years. Age was also stratified into four age groups: 65–79, 80–89, 90–99, 100–. ^c^ Gender was divided into male or female, with females as a reference category; ^d^ Areas were divided into urban areas or rural areas, with rural areas as a reference category; ^e^ Economic independence was divided into “yes (Economic independence)” and “no (Economic dependence)”, with “no (Economic dependence)” as a reference category. ^f^ Household income was divided into “high income” or “low income”, with “low income” as a reference category. ^g^ Education was divided into “at least one year of schooling” or “no schooling”, with “no schooling” as a reference category; ^h^ Marital status was divided into married, widowed, divorced/separated, and never married, with never married as a reference category. ^i^ Smoking was stratified into three groups: non-smoking, past smoking and currently smoking, with non-smoking as a reference category. ^j^ Drinking was stratified into three groups: non-drinking, past drinking, and currently drinking, with non-drinking as a reference category. ^k^ Frequency of fruit intake was stratified into four groups: eating it every day or almost every day, quite often, occasionally, and rarely or never, with rarely or never as a reference category. ^l^ Frequency of vegetable intake was stratified into four groups: eating it every day or almost every day, quite often, occasionally, and rarely or never, with rarely or never as a reference category. ^m^ Basic activities of daily living (ADL) disability was divided into “having ADL disability” or “no ADL disability,” with “no ADL disability” as a reference category. ^n^ Cognitive impairment was divided into “having cognitive impairment” or “no cognitive impairment”, with “no cognitive impairment” as a reference category. ^o^ Systolic blood pressure (SBP) value was a continuous variable. The unit is mmHg. ^p^ Cognitive functions value was a continuous variable. The unit is scores.

**Table 2 ijerph-17-02418-t002:** The scores of cognitive functions by economic independence or dependence among older adults ^a^.

Age (years)	Economic Dependence Group	Cohort1891–1900	Cohort1901–1910	Cohort1911–1920	Cohort1921–1930	Cohort1931–1940	Cohort1941–1950
65–79	Economic dependence				26.1(0.1)	26.9(0.1)	27.8(0.1)
	Economic independence				27.6(0.1)	28.2(0.0)	28.6(0.1)
80–89	Economic dependence			22.5(0.1)	23.7(0.1)	25.1(0.3)	
	Economic independence			25.3(0.2)	26.2(0.1)	27.3(0.3)	
90–99	Economic dependence		17.7(0.1)	18.8(0.1)	20.6(0.4)		
	Economic independence		21.5(0.3)	22.6(0.2)	23.9(0.6)		
100-–	Economic dependence	13.3(0.2)	13.6(0.1)	14.4(0.5)			
	Economic independence	15.6(0.9)	17.2(0.5)	17.3(1.7)			

^a^ Data are expressed as scores mean (standard error) of cognitive functions.

**Table 3 ijerph-17-02418-t003:** The scores of cognitive functions by different income among older adults ^a^.

Age (years)	Household Income Group	Cohort1891–1900	Cohort1901–1910	Cohort1911–1920	Cohort1921–1930	Cohort1931–1940	Cohort1941–1950
65–79	Lower income				26.1(0.1)	27.2(0.1)	28.0(0.1)
	Higher income				27.4(0.1)	27.9(0.0)	28.5(0.1)
80–89	Lower income			22.1(0.1)	23.8(0.1)	25.4(0.3)	
	Higher income			24.3(0.1)	24.8(0.1)	26.1(0.3)	
90–99	Lower income		17.3(0.2)	18.7(0.2)	20.6(0.5)		
	Higher income		19.4(0.2)	20.1(0.1)	21.7(0.4)		
100–	Lower income	12.3(0.3)	13.1(0.2)	13.3(0.7)			
	Higher income	14.8(0.3)	14.6(0.2)	15.6(0.6)			

^a^ Data are expressed as scores mean (standard error) of cognitive functions.

**Table 4 ijerph-17-02418-t004:** The scores of cognitive functions by different education among older adults ^a^.

Age (years)	Education Group	Cohort1891–1900	Cohort1901–1910	Cohort1911–1920	Cohort1921–1930	Cohort1931–1940	Cohort1941–1950
65–79	Lower educated				26.1(0.1)	27.1(0.1)	28.0(0.1)
	Higher educated				27.4(0.1)	28.1(0.0)	28.6(0.1)
80–89	Lower educated			22.1(0.1)	23.7(0.1)	25.3(0.3)	
	Higher educated			24.3(0.1)	25.1(0.1)	26.3(0.3)	
90–99	Lower educated		17.3(0.2)	18.7(0.1)	20.5(0.5)		
	Higher educated		10.4(0.2)	22.4(0.1)	22.0(0.5)		
100–	Lower educated	13.0(0.2)	13.1(0.2)	13.4(0.6)			
	Higher educated	15.7(0.5)	14.8(0.2)	15.9(0.6)			

^a^ Data are expressed as scores mean (standard error) of cognitive functions.

**Table 5 ijerph-17-02418-t005:** Coefficients (95% confidence intervals) from the linear mixed-effects models predicting cognitive functions (Scores).

Variables	Model A		Model B		Model C		Model D	
	Unadjusted	Adjusted	Unadjusted	Adjusted	Unadjusted	Adjusted	Unadjusted	Adjusted
Birth cohorts	3.98(3.88, 4.07)	0.28(0.14, 0.43)	3.68(3.58, 3.77)	0.26(0.12, 0.41)	3.66(3.56, 3.76)	0.27(0.12, 0.41)	3.84(3.74, 3.95)	0.46(0.31, 0.61)
Gender	6.14(5.68, 6.59)	4.16(3.74, 4.58)	5.03(4.57, 5.49)	3.45(3.03, 3.87)	5.03(4.55, 5.52)	3.43(2.99, 3.88)	3.43(2.89, 3.97)	1.97(1.47, 2.46)
Area	1.08(0.68, 1.49)	1.71(1.35, 2.08)	0.37(−0.03, 0.78)	1.26(0.89, 1.63)	0.65(0.22, 1.08)	1.53(1.13, 1.92)	−0.18(−0.63, 0.27)	0.66(0.24, 1.07)
Birth cohorts * Gender	−1.04(−1.16, −0.91)	−0.69(−0.80, −0.57)	−1.04(−1.17, −0.91)	−0.68(−0.80, −0.57)	−1.01(−1.14, −0.88)	−0.67(−0.79, −0.54)	−0.56(−0.71, −0.41)	−0.26(−0.4, −0.12)
Birth cohorts * Area	−0.11(−0.22, −0.005)	−0.21(−0.31, −0.11)	−0.09(−0.20, 0.02)	−0.20−0.30, −0.10)	−0.10(−0.22, 0.01)	−0.24(−0.35, −0.13)	0.15(0.03, 0.28)	0.03(−0.09, 0.14)
Education			1.9(1.71, 2.08)	1.38(1.21, 1.55)	1.82(1.5, 2.14)	1.24(0.95, 1.53)	4.06(3.42, 4.70)	3.1(2.51, 3.68)
Household income			0.76(0.62, 0.90)	0.85(0.72, 0.98)	1.05(0.83, 1.26)	1.10(0.91, 1.3)	1.69(1.27, 2.12)	2.02(1.63, 2.41)
Economic independence			1.33(1.16, 1.51)	0.54(0.38, 0.71)	1.62(1.28, 1.96)	0.56(0.25, 0.87)	4.47(3.71, 5.23)	3.42(2.72, 4.12)
Gender * Economic independence					−0.40(−0.76, −0.04)	−0.21(−0.54, 0.12)	−0.61(−0.97, −0.24)	−0.42(−0.75, −0.09)
Gender*Education					0.13(−0.24, 0.50)	0.20(−0.14, 0.54)	−0.06(−0.43, 0.31)	0.02(−0.32, 0.36)
Gender * Household income					−0.14(−0.41, 0.14)	−0.14(−0.39, 0.11)	−0.05(−0.33, 0.22)	−0.03(−0.28, 0.23)
Area * Economic independence					0.01(−0.33, 0.36)	0.28(−0.04, 0.59)	−0.35(−0.70, 0.01)	−0.08(−0.41, 0.24)
Area * Education					0.02(−0.29, 0.33)	0.05(−0.23, 0.34)	−0.10(−0.41, 0.21)	−0.05(−0.34, 0.23)
Area * Household income					−0.52(−0.79, −0.25)	−0.46(−0.71, −0.20)	−0.49(−0.76, −0.21)	−0.43(−0.68, −0.18)
Birth cohorts * Education							−0.59(−0.74, −0.44)	−0.49(−0.62, −0.35)
Birth cohorts * Household income							−0.20(−0.31, −0.09)	−0.28(−0.39, −0.18)
Birth cohorts * Economic independence							−0.63(−0.79, −0.48)	−0.64(−0.78, −0.50)

*: represents as an interaction between two variables. For example, Birth cohorts * Gender is interpreted birth cohort and gender interaction terms.

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
