# Peer review of "Stronger Increases in Cognitive Functions among Socio-Economically Disadvantaged Older Adults in China: A Longitudinal Analysis with Multiple Birth Cohorts"

_ijerph, 2020, doi:10.3390/ijerph17072418_

Round 1

Reviewer 1 Report

Language is not perfect, but the text is still quite easy to read and understand in most parts. I still think also the small things should be fixed, like “Of especial note”, which should be “Of special note”.

About style: Very many sentences (around 50%?) start with Importantly, Of special note, Furthermore, Additionally, Therefore, Also, Moreover, However et c. Only in the first section, I found four such sentences! In most cases, I think the text would benefit from just omitting those “starters” when they are not really needed.

Here are some other issues:

Importantly, socioeconomic factors (such as low education and income levels) as independent risk factors, may impact cognitive impairment among older adults [8-15]. The potential socioeconomic gradient in health might link with the association between adult socioeconomic status and cognitive function in later life, including exposure to different environments, chronic disease, utilization of the health-care service, and health behaviors [10].

To me, the last sentence is unclear: After reading the first sentence, I expected that the second would add to the first in some way, and then became puzzled about the “link” you propose. But I think you just want to say that the association you mention in the first sentence is in line with the suggested socioeconomic gradient in health? If so, I think you could simplify this a lot to make it more comprehensible? (E.g. like this: Socioeconomic factors (such as education and income levels) may affect with the risk of cognitive impairment among older adults [8-15], in line with the suggested socioeconomic gradient in health (ref).)

Also, after “including…” you enumerate a mix of things that belong to different categories, both to exposures (e.g. environments) and outcomes (chronic diseases). If you want to write anything about this, I suggest you refer to previous studies and specify the associations they found.

The great improvement in nutritional intake [11] and the accessibility and utilization of the health care service [10] among socio-economic disadvantaged groups, which may be more beneficial to the prevention and treatment of cognitive function than before.

First of all, cognitive function is not what needs to be prevented (and treated), but cognitive disorders, or cognitive impairment.

Secondly, the “which” does not belong in the sentence?

Thirdly, if I want to be perhaps too critical, the structure of the sentence is not ideal either, even if I think most readers will understand what you want to say: In a literal sense, you are saying that these improvements may be more beneficial than before. As it is an improvement, and if such an improvement did not happen before, this improvement cannot be more beneficial now than it was before. I guess what you want to say is that these improvements may have contributed to more favourable conditions for prevention and treatment (than before) of cognitive disorders. Or something along those lines? Do you see the difference?

This also seems unclear to me:

However, although socioeconomic disparities in the cognitive impairment of old adults have been studied, most cross-sectional research failed to ascertain the longitudinal change and the cohort effects on cognitive function during the aging process.

This expression: socioeconomic disparities in the cognitive impairment of old people

..looks strange. You mean that the socioeconomic disparities among people with cognitive impairment have been studied? Or perhaps how socioeconomic disparities can affect the risk of cognitive impairment?

The second part also looks strange, because a cross-sectional study cannot evaluate longitudinal changes during the aging process, especially not in cognitive function. It would make more sense to say that most studies have been cross-sectional and thereby have not…(?)

As said, none of these things are very serious, but you could make the text more reader-friendly and prevent confusion by revising wording and grammar as suggested above.

Method:

Respondents with MMSE scores below 18 were defined as having cognitive impairment [22,23].

As this is a study of both cohort effects and of socioeconomic effects, on cognitive function, what is “impaired” may differ between those. It has therefore been suggested to “correct” for this by having different cut-offs for categorization of cognitive impairment; the suggestion has been to use education-based MMSE cut-off points to define cognitive

impairment, ( <18 for persons without formal education, <21 for

with 1-6 years of education, and <25 for persons with more than 6 years of

education. (see e.g. 1.       ZHANG MY, Katzman R, SALMON D, JIN H, CAI GJ, WANG ZY, et al. The Prevalence of Dementia and Alzheimers-Disease in Shanghai, China - Impact of Age, Gender, and Education. Annals of Neurology. 1990 Apr;27(4):428–37.)

As many persons in China lack education, especially among elderly, this adaptation of the MMSE criteria is widely accepted and used in China, and it has also been used previously to define cognitive impairment in the same CLHLS population as you are using (see e.g. Zhu X, Qiu C, Zeng Y, Li J. Leisure activities, education, and cognitive impairment in Chinese older adults: a population-based longitudinal study. International Psychogeriatrics. Cambridge University Press; 2017 May;29(5):727–39.).

In your case, the previously demonstrated bias of using a fixed cut-off for dementia will be even more serious as you are studying “effects” of both cohort and socioeconomic status, i.e. two variables that at the same time are heavily related to level of education.  This seems to be a serious methodological problem in your study, especially since you solely rely on MMSE to determine cognitive impairment.

Unfortunately I was unable to read Tables 2 and 3 properly since the numbers are scrambled across the page. Please make sure that the pdf comes out right when you submit the next version of the manuscript. But from what I think I can read there are minuses everywhere:

You need to also define your outcome variables: What do you mean by cohort trend and “decrease in the cohort trend”? Decrease in relation to what? Do you mean that the effect of cohort gets smaller with later cohorts, compared to the first one (1906-1918), irrespective of age? If the first cohort is a reference, how could also this cohort have a negative value (in relation to what?) Does the – (minus) indicate that the cohort had less of an effect on cognitive impairment, as defined here? I simply don´t understand this. When I read the Discussion I get the impression that my confusion could be related to the use of language: I don’t think you mean that the trend is decreasing, but that there is a trend, that it is strong, and that you mean that this trend means that that the prevalence of cognitive impairment is decreasing with later cohorts? (If you say that a trend is decreasing, it means that the trend was stronger before than it is now, which is a different thing.) Could you please revise the language to express this in a clearer way?

The same thing with “the negative effect of education on cognitive impairment”. How can education have a negative effect on cognitive impairment? From the text it seems to be the opposite, that absence of education has a negative effect? Or do you mean with “negative” that education decreases the risk of cognitive impairment? (And thus that education has a positive effect?) If so, again you need to revise the language to express this in a clearer way.)

Finally, I think you need to explain better what is meant by the minuses in the tables. What do these numbers stand for? And minus in relation to what?

In summary, at this point these are my biggest concerns:

  1. The use of MMSE as the only estimate for cognitive impairment, and that in addition cut-offs do not take education differences into account. This could, especially in your case, introduce a heavy bias, exaggerating the number of people with cognitive impairment in earlier cohorts. At least, you should also provide results with this common and generally recommended way to use MMSE cut-offs in China, a method that was previously used on the same sample you have investigated.
  2. Terminology should be improved to become more explicit in several instances, and I think you especially need to clarify/revise the terminology around the concepts “decrease in the cohort trend” and “the negative effect of education”.
  3. The tables need to be cleaned up so that numbers and letters are aligned, and you also need to describe the main results in a way that harmonizes better with the tables. For example, I do not understand how all the cohorts could have a minus value (minus in relation to what, there is no reference? Or is it a measure of incidence of cognitive impairment from baseline to follow-up?) I would also appreciate if you could explain in common language how the coefficients could be so high, rather than just presenting the formulas?

Author Response

Dear Editors and Reviewers,

Thanks for these comments. We agree with these suggestions, which might be of great help to improve the quality of our manuscript.

We have revised the manuscript and responded, point by point to, the comments as listed below. We would like to re-submit this revised manuscript to IJERPH, and hope it is acceptable for publication in the journal. The primary corrections in the paper and the response to the reviewer's comments are as follows:

Replies to the Reviewer Comments and Suggestions:

Reviewer 1##:

1. Language is not perfect, but the text is still quite easy to read and understand in most parts. I still think also the small things should be fixed, like "Of especial note", which should be "Of special note".

Answer:
Thank you very much for pointing out these problems. We have sought out an English editor to polish the English language of our manuscript. We have simplified this sentence, so we deleted this inappropriate word. "China has rapidly become an aging society, where the number of people aged 65 and older will increase from 6.9% of the population in 2000 to 22.7% in 2050 [1, 2]."

2. About style: Very many sentences (around 50%?) start with Importantly, Of special note, Furthermore, Additionally, Therefore, Also, Moreover, However et c. Only in the first section, I found four such sentences! In most cases, I think the text would benefit from just omitting those "starters" when they are not really needed.

Answer:
According to the suggestion, we have omitted those "starters" when they are not needed. E.g., "Cognitive aging has been defined as "a process of gradual, ongoing, yet highly variable changes in cognitive functions that occur as people get older" [4].", "To gain deeper insights into the development of cognitive functions, it is necessary to study the longitudinal trend in cognitive functions among different socio-economic older adults.", "Nearly two-thirds of the current older adults in China were involved in farming and agricultural production when they were young, preventing them from having a stable income in their later life [17]. Economic independence may reflect the financial situation and income sources of older adults in China [18]."

3. Importantly, socio-economic factors (such as low education and income levels) as independent risk factors, may impact cognitive impairment among older adults [8-15]. The potential socio-economic gradient in health might link with the association between adult socio-economic status and cognitive function in later life, including exposure to different environments, chronic disease, utilization of the health-care service, and health behaviors [10].
To me, the last sentence is unclear: After reading the first sentence, I expected that the second would add to the first in some way, and then became puzzled about the "link" you propose. But I think you just want to say that the association you mention in the first sentence is in line with the suggested socio-economic gradient in health? If so, I think you could simplify this a lot to make it more comprehensible? (E.g. like this: Socio-economic factors (such as education and income levels) may affect with the risk of cognitive impairment among older adults [8-15], in line with the suggested socio-economic gradient in health (ref).)
 Also, after "including…" you enumerate a mix of things that belong to different categories, both to exposures (e.g. environments) and outcomes (chronic diseases). If you want to write anything about this, I suggest you refer to previous studies and specify the associations they found.

Answer:
We thank the reviewer for raising this problem to help us improve the manuscript. According to the suggestions, we simplify it to make it more comprehensible. "Socio-economic factors (such as education and income levels) may affect the risk of cognitive decline among older adults [8–15], in line with the suggested socio-economic gradient in health." We also deleted this sentence, "including exposure to different environments, chronic disease, utilization of the health-care service, and health behaviors."

4.The great improvement in nutritional intake [11] and the accessibility and utilization of the health care service [10] among socio-economic disadvantaged groups, which may be more beneficial to the prevention and treatment of cognitive function than before.
First of all, cognitive function is not what needs to be prevented (and treated), but cognitive disorders, or cognitive impairment.
Secondly, the "which" does not belong in the sentence?
Thirdly, if I want to be perhaps too critical, the structure of the sentence is not ideal either, even if I think most readers will understand what you want to say: In a literal sense, you are saying that these improvements may be more beneficial than before. As it is an improvement, and if such an improvement did not happen before, this improvement cannot be more beneficial now than it was before. I guess what you want to say is that these improvements may have contributed to more favourable conditions for prevention and treatment (than before) of cognitive disorders. Or something along those lines? Do you see the difference?

Answer:
As the reviewer noted, this sentence has caused misunderstandings and has not been clearly explained. According to the reviewer's explanation, we understand the difference. We changed it, "There were considerable improvements in nutritional intake [11] and the accessibility and utilization of the health-care service [10] among disadvantaged socio-economic groups. These improvements may have contributed to more favorable conditions for the prevention and treatment (than previously) of cognitive decline."

5. This also seems unclear to me:
However, although socio-economic disparities in the cognitive impairment of old adults have been studied, most cross-sectional research failed to ascertain the longitudinal change and the cohort effects on cognitive function during the aging process.
This expression: socio-economic disparities in the cognitive impairment of old people
..looks strange. You mean that the socio-economic disparities among people with cognitive impairment have been studied? Or perhaps how socio-economic disparities can affect the risk of cognitive impairment?
The second part also looks strange, because a cross-sectional study cannot evaluate longitudinal changes during the aging process, especially not in cognitive function. It would make more sense to say that most studies have been cross-sectional and thereby have not…(?)
 As said, none of these things are very serious, but you could make the text more reader-friendly and prevent confusion by revising wording and grammar as suggested above.

Answer:
As the reviewer noted, these sentences have not been clearly explained. According to the suggestion, we changed it into "However, some studies showed that older adults with lower socio-economic status have weaker cognitive functions. Most studies were cross-sectional and did not ascertain the longitudinal change and the cohort effects on cognitive function during the aging process."
Thank you very much for pointing out these words and grammar problems. We have sought out an English editor to polish the English language of our manuscript.

6. Method:
Respondents with MMSE scores below 18 were defined as having cognitive impairment [22,23].
As this is a study of both cohort effects and of socio-economic effects, on cognitive function, what is "impaired" may differ between those. It has therefore been suggested to "correct" for this by having different cut-offs for categorization of cognitive impairment; the suggestion has been to use education-based MMSE cut-off points to define cognitive
impairment, ( <18 for persons without formal education, <21 for
with 1-6 years of education, and <25 for persons with more than 6 years of
education. (see e.g. 1.       ZHANG MY, Katzman R, SALMON D, JIN H, CAI GJ, WANG ZY, et al. The Prevalence of Dementia and Alzheimers-Disease in Shanghai, China - Impact of Age, Gender, and Education. Annals of Neurology. 1990 Apr;27(4):428–37.)
As many persons in China lack education, especially among elderly, this adaptation of the MMSE criteria is widely accepted and used in China, and it has also been used previously to define cognitive impairment in the same CLHLS population as you are using (see e.g. Zhu X, Qiu C, Zeng Y, Li J. Leisure activities, education, and cognitive impairment in Chinese older adults: a population-based longitudinal study. International Psychogeriatrics. Cambridge University Press; 2017 May;29(5):727–39.).
In your case, the previously demonstrated bias of using a fixed cut-off for dementia will be even more serious as you are studying "effects" of both cohort and socio-economic status, i.e. two variables that at the same time are heavily related to level of education.  This seems to be a serious methodological problem in your study, especially since you solely rely on MMSE to determine cognitive impairment.

Answer:
We thank the reviewer for raising this important issue to help us improve the manuscript. Based on this literature [Zhu X, Qiu C, Zeng Y, Li J. Leisure activities, education, and cognitive impairment in Chinese older adults: a population-based longitudinal study. International Psychogeriatrics. Cambridge University Press; 2017 May;29(5):727–39.] and reviewer 2 and 4's suggestions, we have revised to use the education-based MMSE cut-off point [23] for the definition of cognitive impairment in our study. "A large proportion (60.3%) of the older adults in this study had no formal schooling. Therefore, in this study, the education-based MMSE cut-off point was used to define cognitive impairment. This cut-off-point is widely accepted for screening for cognitive impairment in the older adult population with less education [23]. An MMSE score < 18 for participants with no formal education had, an MMSE score of < 21 for those with 1 to 6 years of education had, or an MMSE score of < 25 for those with more than 6 years of education was defined as indicating cognitive impairment [23]."
Also, other reviewers pointed out this problem at the same time and suggested that cognitive functions (scores) be used as a continuous variable for analysis. Therefore, cognitive function is used as a dependent variable in the analysis model after modification.

7.Unfortunately I was unable to read Tables 2 and 3 properly since the numbers are scrambled across the page. Please make sure that the pdf comes out right when you submit the next version of the manuscript. But from what I think I can read there are minuses everywhere:

Answer:
We have changed the Table into one page and hope its pdf comes out right so that you can read these Tables properly.

We appreciate for Editors/Reviewers’ warm work earnestly and hope that the correction will meet with approval.

Once again, thank you very much for your comments and suggestions.

With kindest regards,

Yours Sincerely

Fan Yang

Reviewer 2 Report

Yang et al., analyzed a longitudinal study performed by Chinese Longitudinal Healthy Longevity Survey on 14,929 older adults over 10 years in order to predict which socio-economic factors are affecting the cognitive impairment in such adults.

The topic of research is interesting, the manuscript is a very well written, the study objective is clear and all variables have been carefully defined and examined. I have only one minor concern about this study:

  • Did the authors use some exclusion criteria (eg. Individuals with some diagnosed neurodegenerative disease, etc. )? Please, specify it in methods.

Author Response

Dear Editors and Reviewers,

Thanks for these comments. We agree with these suggestions, which might be of great help to improve the quality of our manuscript.

We have revised the manuscript and responded, point by point to, the comments as listed below. We would like to re-submit this revised manuscript to IJERPH, and hope it is acceptable for publication in the journal. The primary corrections in the paper and the response to the reviewer's comments are as follows:

Replies to the Reviewer Comments and Suggestions:

Reviewer 2##

Yang et al., analyzed a longitudinal study performed by Chinese Longitudinal Healthy Longevity Survey on 14,929 older adults over 10 years in order to predict which socio-economic factors are affecting the cognitive impairment in such adults.

The topic of research is interesting, the manuscript is a very well written, the study objective is clear and all variables have been carefully defined and examined. I have only one minor concern about this study:

Did the authors use some exclusion criteria (eg. Individuals with some diagnosed neurodegenerative disease, etc. )? Please, specify it in methods.

Answer:

Thank you very much for pointing out the important issues.

Because the CLHLS surveyed the population of older adults (over 65 years old), and all cognitive questions (MMSE) must be answered by themselves, no one else could answer for them. This study did not use certain criteria to exclude patients with certain diseases.

Based on the problem you pointed out, we specifically pointed out and explained it to the reader in the limitations section of this study. “Individuals who had been diagnosed with the neurodegenerative disease were not excluded from the CLHLS.”

We appreciate for Editors/Reviewers’ warm work earnestly and hope that the correction will meet with approval.

Once again, thank you very much for your comments and suggestions.

With kindest regards,

Yours Sincerely

Fan Yang

Reviewer 3 Report

The manuscript lacks line numbers, so it is not so easy to refer to specific text passages. Probably copy editor forgot to add these. I have used quotes ("") from the manuscript to refer to specific places.

I am happy to see more research based on large Chinese surveys being studied and published in English.

"people aged 60 and older will reach 30% of the population in 2050 [1], and those over 65 years will increase from 6.9% in 2000 to 22.7% in 2050. Of especial note, among the older population, the percentage of the oldest-old people aged 80 and older will exceed 30% in 2050 [2]."

Seems incoherent. Maybe make a plot to show these projections and past trends.

The term "oldest-old" is very odd. I see that it's a standard term, but you should define it clearly since most readers will not be aware of its meaning. I see that other sources define it as 85+, but you seem to use 80+.

"Moreover, there is a scarcity of research assessing the longitudinal trends [11-13] and cohort effects [11,14] on cognitive disparities among different socioeconomic groups – subjects have all come from developed countries, particularly the United States, with little attention to developing countries."

The authors are right that most studies with such designs are from western developed countries, but not all of them.

"Besides, there is currently no research analyzing socioeconomic effects in cognitive impairment among older adults of different genders and living in different areas."

In China.

"across 22 provinces"

You may want to tell the reader how many provinces there are in total, and which percentage this covers. Probably they sampled from the most populous provinces, so you can calculate the weighted percent coverage of provinces.

"1998 and carried out the subsequent five rounds of data collection in 2000, 2002, 2005, 2008, and 2012"

Why did it stop? 2012 is 8 years ago, seems like one should do another wave.

"Cognitive function was measured using the Chinese version of the Mini Mental State
Examination (MMSE)"

Could you remind the reader how many items the test has in total? How many for each of the 4 subscales?

"Marital status was classified as either “married” or “others.”"

Seems like it would be useful to follow standard codings, e.g.: married, dirvorced/separated, widowed.

"Household income was classified based on the median of per capita net income of urban and rural areas for each survey year: “high income” or “low income.”"

So this is actually not own household income, but that of the neighborhood or area. Please do not discretize data is possible. This throws away information. Always use continuous variables if possible.

Table 1: I think it would be better to compute the mean MMSE score than the proportion of people with scores < 18 You can do both.

Your outcome is a binary variable, but you don't say whether you are using logistic regression or OLS. This is confusing. The equations in A-D suggest you are using OLS.

You should plot some of the model predictions to see if the estimates are sensible. I don't know how one can do this with STATA, but one can easily do this with R. See e.g. ggeffects package https://joss.theoj.org/papers/10.21105/joss.00772. You can view some of the model predictions and see if they produce realistic forecasts.

You modeled the birth cohorts as fixed effects it seems, instead of a continuous variable. This will prevent you from making predictions about the future, e.g., about the 1946-1955 cohort.

"Lastly, the MMSE as a sole measure of cognitive impairment is inadequate because the MMSE has limited diagnostic utility in this setting."

The MMSE is basically a poor or quick IQ test. If one used a longer test, the results would be stronger. It may be possible to score the test using machine learning to improve the results.

"It allows for the development of effective public intervention targeting this disadvantaged population, and more considerable efforts should be put into aiding females and those who live in rural areas in order to reduce cognitive impairment."

This is not supported by the regressions. You cannot assume these statistical associations are causal.

Author Response

Dear Editors and Reviewers,

Thanks for these comments. We agree with these suggestions, which might be of great help to improve the quality of our manuscript.

We have revised the manuscript and responded, point by point to, the comments as listed below. We would like to re-submit this revised manuscript to IJERPH, and hope it is acceptable for publication in the journal. The primary corrections in the paper and the response to the reviewer's comments are as follows:

Replies to the Reviewer Comments and Suggestions:

Reviewer 3##

  1. The manuscript lacks line numbers, so it is not so easy to refer to specific text passages. Probably copy editor forgot to add these. I have used quotes ("") from the manuscript to refer to specific places.

I am happy to see more research based on large Chinese surveys being studied and published in English.

"people aged 60 and older will reach 30% of the population in 2050 [1], and those over 65 years will increase from 6.9% in 2000 to 22.7% in 2050. Of especial note, among the older population, the percentage of the oldest-old people aged 80 and older will exceed 30% in 2050 [2]."

Seems incoherent. Maybe make a plot to show these projections and past trends.

Answer:

Based on the suggestions, we have simplified the sentences for clarity. “China has rapidly become an aging society, where the number of people aged 65 and older will increase from 6.9% of the population in 2000 to 22.7% in 2050 [1, 2].”

  1. The term "oldest-old" is very odd. I see that it's a standard term, but you should define it clearly since most readers will not be aware of its meaning. I see that other sources define it as 85+, but you seem to use 80+.

Answer:

We thank the reviewer for raising this problem to help us improve the manuscript. According to the suggestion, we just used older adults aged 80 and above to make it more clearly in the text.

  1. "Moreover, there is a scarcity of research assessing the longitudinal trends [11-13] and cohort effects [11,14] on cognitive disparities among different socioeconomic groups – subjects have all come from developed countries, particularly the United States, with little attention to developing countries."

The authors are right that most studies with such designs are from western developed countries, but not all of them.

"Besides, there is currently no research analyzing socioeconomic effects in cognitive impairment among older adults of different genders and living in different areas."

Answer:

We thank the reviewer for pointing out this. We have changed it, “Many research studies assessing longitudinal trends in cognitive functions came from developed countries, particularly the United States.” We also have deleted this sentence "Besides, there is currently no research analyzing socioeconomic effects in cognitive impairment among older adults of different genders and living in different areas." in the text.

  1. In China.

"across 22 provinces"

You may want to tell the reader how many provinces there are in total, and which percentage this covers. Probably they sampled from the most populous provinces, so you can calculate the weighted percent coverage of provinces.

"1998 and carried out the subsequent five rounds of data collection in 2000, 2002, 2005, 2008, and 2012"

Why did it stop? 2012 is 8 years ago, seems like one should do another wave.

Answer:

Based on the suggestions, we have changed it to “The CLHLS involved one of the most extensive samples of older adults from 631 counties/cities in 22 provinces (China has a total of 31 provincial-level regions), covering up to 85% of the total Chinese population in China[17,19-20].”

Moreover, according to the 2014 data released by CLHLS, we include the data of 2014 CLHLS and re-update our analysis.

  1. "Cognitive function was measured using the Chinese version of the Mini Mental State

Examination (MMSE)"

Could you remind the reader how many items the test has in total? How many for each of the 4 subscales?

Answer:

Based on the suggestions, we have changed it to “Cognitive function was measured using the Chinese version of the Mini Mental State Examination (MMSE) adapted from the scale developed by Folstein and colleagues [21, 22], evaluating five aspects of functioning: orientation, reaction, attention and calculation, recall, and language [15]. The Chinese version of the MMSE we used fits the cultural and socio-economic background in China, and the reliability was high (Cronbach’s a = 0.94) for the current samples of Chinese older adults [15]. This MMSE consists of 24 questions covering orientation (six questions), reaction (three questions), attention and calculation (six questions), recall (three questions), and language (six questions) [15, 23].”

  1. "Marital status was classified as either “married” or “others.”"

Seems like it would be useful to follow standard codings, e.g.: married, dirvorced/separated, widowed.

Answer:

Based on the suggestions, we have changed it to “Marital status was classified as married, widowed, divorced/separated, or never married.”

  1. "Household income was classified based on the median of per capita net income of urban and rural areas for each survey year: “high income” or “low income.”"

So this is actually not own household income, but that of the neighborhood or area. Please do not discretize data is possible. This throws away information. Always use continuous variables if possible.

Answer:

Because in the CLHLS, if the income per capita of a household last year was more than 100,000 Chinese yuan, the CLHLS coded 99998. So this study did not use household income as a continuous variable because the CLHLS did not code the exact value of income above 100,000 Chinese yuan. Household income was classified based on the median of per capita net income for each survey year: “high income” or “low income.”

8.Table 1: I think it would be better to compute the mean MMSE score than the proportion of people with scores < 18 You can do both.

Your outcome is a binary variable, but you don't say whether you are using logistic regression or OLS. This is confusing. The equations in A-D suggest you are using OLS.

You should plot some of the model predictions to see if the estimates are sensible. I don't know how one can do this with STATA, but one can easily do this with R. See e.g. ggeffects package https://joss.theoj.org/papers/10.21105/joss.00772. You can view some of the model predictions and see if they produce realistic forecasts.

Answer:

Based on the suggestions of you and the fourth reviewer, we included cognitive function as a continuous independent variable in the analysis model and indicated that the study used the linear mixed-effects model for analysis. “Linear mixed-effects models are especially useful in longitudinal studies [27] and were used to identify cohort trends in cognitive functions among different socio-economic older adults.”

Also, although we have not used this model to show such predictions, we provide the cognitive function (Table 2-4) and cognitive impairment (Figure 1) by different cohorts (and different age groups). We will try to learn R for the data analysis.

  1. You modeled the birth cohorts as fixed effects it seems, instead of a continuous variable. This will prevent you from making predictions about the future, e.g., about the 1946-1955 cohort.

Answer:

We thank the reviewer for raising this issue to help us improve the manuscript. We have changed it to “Birth cohorts was a continuous variable: cohort 1891–1900, cohort 1901–1910, cohort 1911–1920, cohort 1921–1930, cohort 1931–1940, and cohort 1941–1950.”, “When other socio-economic factors were adjusted, cognitive functions also increased significantly by 0.28 scores per birth cohort (coefficient [95% confidence intervals]: 0.28 scores [0.14, 0.43]; Table 5, adjusted model A).”

  1. "Lastly, the MMSE as a sole measure of cognitive impairment is inadequate because the MMSE has limited diagnostic utility in this setting."

The MMSE is basically a poor or quick IQ test. If one used a longer test, the results would be stronger. It may be possible to score the test using machine learning to improve the results.

Answer:

We thank the reviewer for raising this issue. At the same time, the first and fourth reviewers pointed out this important issue. According to their suggestions, we have revised into use the education-based MMSE cut-off point [23] for the definition of cognitive impairment in our study. “A large proportion (60.3%) of the older adults in this study had no formal schooling. Therefore, in this study, the education-based MMSE cut-off point was used to define cognitive impairment. This cut-off-point is widely accepted for screening for cognitive impairment in the older adult population with less education [23]. An MMSE score < 18 for participants with no formal education had, an MMSE score of < 21 for those with 1 to 6 years of education had, or an MMSE score of < 25 for those with more than 6 years of education was defined as indicating cognitive impairment [23].”

Also, other reviewers suggested that cognitive functions (scores) be used as a continuous variable for analysis. Therefore, cognitive function was used as the dependent variable in the analysis model after modification.

11."It allows for the development of effective public intervention targeting this disadvantaged population, and more considerable efforts should be put into aiding females and those who live in rural areas in order to reduce cognitive impairment."

This is not supported by the regressions. You cannot assume these statistical associations are causal.

Answer:

According to the suggestion, we have deleted it in the text.

We appreciate for Editors/Reviewers’ warm work earnestly and hope that the correction will meet with approval.

Once again, thank you very much for your comments and suggestions.

With kindest regards,

Yours Sincerely

Fan Yang

Reviewer 4 Report

This is an interesting and very well-written manuscript about a current topic in aging research, how does cognitive trajectories are impacted by non-biological factors, such as education and economic status. I have some suggestions:

  • Maybe argumentation in the last paragraph of the introduction on why the objective addressed by the authors is important, could be reduced. In addition, authors comment that there are no longitudinal studies on cognitive impairment in the least favored populations; however, it would be important to mention that studies such as 10/66 and SAGE, are precisely to address this issue and certainly have explored the longitudinal nature of cognitive decline. 
  • I am afraid the main flaw of the study and a very difficult one to address is the decision on how to define cognitive impairment. There is an appropriate description in the introduction, that cognitive impairment could represent an intermediate state between normal aging and dementia; however, it is not clear what happened to those individuals who acquired the dementia diagnosis in the follow-up. Where they not followed-up any more? Where individuals with dementia excluded from the study? Moreover, the cut-off value for the MMSE is referred from a population of Mexican Americans, and that was the definition of cognitive impairment. This might not be appropriate, and strongly recommend to use values for the Chinese population. Authors acknowledge the limitations of the MMSE in assessing cognition, however they do not provide any further argumentation on its strengths in testing cognitive problems, nor if the adapted and validated version they used, has appropriate diagnostic characteristics in order to detect cognitive impairment in Chinese older adults. Furthermore, abundant literature on cognitive impairment refers to the adjustment for at least education; other advise on using sex and even age into the adjustment of the cut-off values in order to define if an individual has cognitive impairment. Therefore the results on education do not surprise at all. 
  • In order to tackle this problem, a possibly better approach would have been to use the continuous score of MMSE, this would have provided with a wider spectrum of what the cognitive decline looks like, and avoid black/white responses in a phenomenon that in nature is continuous. 
  • Regarding adjustment, depressive symptoms and sensorial impairments were not included for adjustment; and in the side of lifestyle behaviors, alcohol and tobacco use were included, but not a word on physical activity and nutrition. Would suggest to further consider adjusting for these factors. 
  • Also, besides SBP, no other chronic disease was considered in the modeling. 
  • How does the participation of women in the economy could affect the impact of economic dependency? Could authors please provide more information and arguments on this matter?

Author Response

Dear Editors and Reviewers,

Thanks for these comments. We agree with these suggestions, which might be of great help to improve the quality of our manuscript.

We have revised the manuscript and responded, point by point to, the comments as listed below. We would like to re-submit this revised manuscript to IJERPH, and hope it is acceptable for publication in the journal. The primary corrections in the paper and the response to the reviewer's comments are as follows:

Replies to the Reviewer Comments and Suggestions:

Reviewer 4##

This is an interesting and very well-written manuscript about a current topic in aging research, how does cognitive trajectories are impacted by non-biological factors, such as education and economic status. I have some suggestions:

  1. Maybe argumentation in the last paragraph of the introduction on why the objective addressed by the authors is important, could be reduced. In addition, authors comment that there are no longitudinal studies on cognitive impairment in the least favored populations; however, it would be important to mention that studies such as 10/66 and SAGE, are precisely to address this issue and certainly have explored the longitudinal nature of cognitive decline.

Answer:

Based on the suggestions, we have changed it to “However, some studies showed that older adults with lower socio-economic status have weaker cognitive functions. Most studies were cross-sectional and did not ascertain the longitudinal change and the cohort effects on cognitive function during the aging process. A longitudinal study with multiple birth cohorts has the clear advantage of exploring complex longitudinal trends and cohort effects in cognitive functions during the aging process and providing precise explanations of socio-economic effects [11–14]. Many research studies assessing longitudinal trends in cognitive functions came from developed countries, particularly the United States. Moreover, owing to data limitations, most of the studies have typically investigated subjects younger than 80 years.”

  1. I am afraid the main flaw of the study and a very difficult one to address is the decision on how to define cognitive impairment. There is an appropriate description in the introduction, that cognitive impairment could represent an intermediate state between normal aging and dementia; however, it is not clear what happened to those individuals who acquired the dementia diagnosis in the follow-up. Where they not followed-up any more? Where individuals with dementia excluded from the study? Moreover, the cut-off value for the MMSE is referred from a population of Mexican Americans, and that was the definition of cognitive impairment. This might not be appropriate, and strongly recommend to use values for the Chinese population. Authors acknowledge the limitations of the MMSE in assessing cognition, however they do not provide any further argumentation on its strengths in testing cognitive problems, nor if the adapted and validated version they used, has appropriate diagnostic characteristics in order to detect cognitive impairment in Chinese older adults. Furthermore, abundant literature on cognitive impairment refers to the adjustment for at least education; other advise on using sex and even age into the adjustment of the cut-off values in order to define if an individual has cognitive impairment. Therefore the results on education do not surprise at all.

Answer:

We thank the reviewer for raising this important problem. At the same time, the first and third reviewers also pointed out this important issue. According to the suggestions, we have revised to use the education-based MMSE cut-off point [23] for the definition of cognitive impairment in our study. “A large proportion (60.3%) of the older adults in this study had no formal schooling. Therefore, in this study, the education-based MMSE cut-off point was used to define cognitive impairment. This cut-off-point is widely accepted for screening for cognitive impairment in the older adult population with less education [23]. An MMSE score < 18 for participants with no formal education had, an MMSE score of < 21 for those with 1 to 6 years of education had, or an MMSE score of < 25 for those with more than 6 years of education was defined as indicating cognitive impairment [23].” [Zhu X, Qiu C, Zeng Y, Li J. Leisure activities, education, and cognitive impairment in Chinese older adults: a population-based longitudinal study. International Psychogeriatrics. Cambridge University Press; 2017 May;29(5):727–39.]

Because the CLHLS is not limited to study cognitive impairment in older adults, the older adults are followed as much as possible. [17.    Gu, D.; Zeng, Y. Data assessment of the Chinese Longitudinal Healthy Longevity Survey. In: Zeng, Y., Xiao, Z., Zhang, C., editors. Determinants of Healthy Longevity of the Oldest-Old in China. Beijing: Peking University Press, 2004.] The CLHLS surveyed the population of older adults (over 65 years old), and all cognitive questions must be answered by themselves, no one else could answer for them. This study did not use specific criteria to exclude patients with certain diseases.  Based on the problem you pointed out, we specifically pointed out and explained it to the reader in the limitations section of this study. “Individuals who had been diagnosed with neurodegenerative disease or dementia were not excluded from the CLHLS.”

  1. In order to tackle this problem, a possibly better approach would have been to use the continuous score of MMSE, this would have provided with a wider spectrum of what the cognitive decline looks like, and avoid black/white responses in a phenomenon that in nature is continuous.

Answer:

We thank the reviewer for raising this issue to help us improve the manuscript. According to the suggestions, cognitive functions (scores) was used as a continuous variable in our analysis. Therefore, cognitive function was used as the dependent variable in the analysis model after modification.

  1. Regarding adjustment, depressive symptoms and sensorial impairments were not included for adjustment; and in the side of lifestyle behaviors, alcohol and tobacco use were included, but not a word on physical activity and nutrition. Would suggest to further consider adjusting for these factors. Also, besides SBP, no other chronic disease was considered in the modeling.

Answer:

According to the suggestions, we add “frequency of fruit intake, frequency of vegetable intake, and ADL disability” as new adjusted variables in analysis models. “Frequency of fresh fruit intake was stratified into four groups: eating it every day or almost every day, quite often, occasionally, and rarely or never. Frequency of fresh vegetable intake was also stratified into four groups: eating it every day or almost every day, quite often, occasionally, and rarely or never. Functional disability was assessed by self-report using the Katz questionnaire for basic activities of daily living (ADL). In the clinical and epidemiological fields, this evaluation tool is used to assess functional independence [19]. ADL was evaluated by six daily self-care tasks: bathing, eating, getting in and out of bed, dressing, going to the toilet, and maintaining control of urine and feces. Respondents who reported any assistance in any items were categorized as ADL disability, whereas respondents who completed all items without any help were categorized as independent of ADL disability [19].”

In addition, we point out these problems to readers in research limitations.“ We adjusted the variables (such as age, marital status, smoking, drinking, frequency of fruit intake, frequency of vegetable intake, SBP, and ADL disability) in the models during the data analysis. However, some other impact factors, such as physical activity, medical history, and sensorial impairments, were not collected in the CLHLS and cannot be added to these analysis models.”

  1. How does the participation of women in the economy could affect the impact of economic dependency? Could authors please provide more information and arguments on this matter?

Answer:

As we re-adopted cognitive functions as the continuous dependent variable and the birth cohort as the continuous independent variable, the analysis results showed “The magnitude of differences in cognitive functions between older adults with higher and lower incomes was reflected in scores lower by 0.46 among those living in urban areas than among those living in rural areas (coefficient [95% CI]: -0.46 scores [-0.71, -0.20]; the interaction between household income and area in Table 5, adjusted model C).”

Therefore, we only analyze and explain this difference in the discussion. Since we did not provide more information and arguments in the study results clearly in the previous manuscript, we have explained in detail in this section while revising. “ China has a unique urban–rural structure where economic development and stratification systems differ from those in western developed countries, as nearly two-thirds of the current old adults in China were involved in farming and agricultural production when they were young [17, 34–36]. Older adults in rural areas were generally socio-economically disadvantaged with limited social participation and less cognitively stimulating environments relative to those in urban areas [13,14,33,37]. Rural residents with lower income levels may have lower resources and opportunities for social engagement and interaction to prevent their cognitive decline than their counterparts. Rural areas also lack economic resources. Deprived neighborhoods may result in fewer resources for health-service utilization [11–14,16,34], which otherwise may have posed barriers to against cognitive decline among impoverished rural residents [38]. It may be one reason for our finding of a more significant magnitude of income-related disparity in cognitive functions among rural-area older adults in China.”

We appreciate for Editors/Reviewers’ warm work earnestly and hope that the correction will meet with approval.

Once again, thank you very much for your comments and suggestions.

With kindest regards,

Yours Sincerely

Fan Yang

Round 2

Reviewer 3 Report

I am fairly happy with this now. It has improved a lot since last. Two minor issues.

First, the cohorts are not modeled as continuous variables (this means, use birth year as a predictor). At least, I don't think so. When the authors write it increases by birth cohort, I think they are just showing that it gives up every 10 years of birth year. This accomplishes the same thing, but in a suboptimal way (data are discretized). In general, it is bad to discretize (to bin) data. This is a good general reference. https://onlinelibrary.wiley.com/doi/abs/10.1002/sim.2331

Second, the figure is quite unclear. Please use a higher resolution.

Author Response

Dear Editors and Reviewers,

Thanks for these comments. We agree with these suggestions, which might be of great help to improve the quality of our manuscript.

We have revised the manuscript and responded, point by point to, the comments as listed below. We want to re-submit this revised manuscript to IJERPH, and hope it is acceptable for publication in the journal. The primary corrections in the paper and the response to the reviewer's comments are as follows:

Replies to the Reviewer Comments and Suggestions:

Reviewer 3##:

I am fairly happy with this now. It has improved a lot since last. Two minor issues.

  1. First, the cohorts are not modeled as continuous variables (this means, use birth year as a predictor). At least, I don't think so. When the authors write it increases by birth cohort, I think they are just showing that it gives up every 10 years of birth year. This accomplishes the same thing, but in a suboptimal way (data are discretized). In general, it is bad to discretize (to bin) data. This is a good general reference. https://onlinelibrary.wiley.com/doi/abs/10.1002/sim.2331

Answer:

We strongly agree with this opinion and have learned from this good general reference . We marked the sample to six birth cohorts (1891–1900, cohort 1901–1910, cohort 1911–1920, cohort 1921–1930, cohort 1931–1940, and cohort 1941–1950) and included this variable (birth cohorts) into the model as a continuous variable to analysis.

We will also refer to these suggestions and this literature in our future data analysis and paper writing( including continuous variables into the model as far as possible).

  1. Second, the figure is quite unclear. Please use a higher resolution.

Answer:

According to the suggestion, we have provided a higher resolution of the Figure.

We appreciate for Editors/Reviewers' warm work earnestly and hope that the correction will meet with approval.

Once again, thank you very much for your comments and suggestions.

With kindest regards,

Yours Sincerely

Fan Yang

Reviewer 4 Report

Worth to be published.

Author Response

Dear Editors and Reviewers,

Thanks for these comments. We want to re-submit this revised manuscript to IJERPH, and hope it is acceptable for publication in the journal. The primary corrections in the paper and the response to the reviewer's comments are as follows:

Replies to the Reviewer Comments and Suggestions:

Reviewer 4##:

  1. Worth to be published.

Answer:

Thank the reviewer for putting forward many suggestions to help us improve our manuscript in the last review.

We appreciate for Editors/Reviewers' warm work earnestly and hope that the correction will meet with approval.

Once again, thank you very much for your comments and suggestions.

With kindest regards,

Yours Sincerely

Fan Yang
